# Cimicids of Medical and Veterinary Importance

**DOI:** 10.3390/insects14040392

**Published:** 2023-04-17

**Authors:** Fatima Zohra Hamlili, Jean Michel Bérenger, Philippe Parola

**Affiliations:** 1IHU-Méditerranée Infection, 19–21 Boulevard Jean Moulin, 13005 Marseille, France; 2Department of VITROME, Aix Marseille Univ, IRD, AP-HM Assistance Publique-Hôpitaux de Marseille, SSA, 13005 Marseille, France

**Keywords:** cimicids, bed bugs, bat bugs, swallow bugs, poultry bugs, bites, human health, animal health

## Abstract

**Simple Summary:**

Cimicids are obligate bloodsuckers of animals and vertebrate. Some species have humans as primary hosts and are currently called “bed bugs”. Other cimicid species related to birds and bats can also bite humans, especially if these animals have their nests near human dwellings. In this paper, we summarize all the cases found in the literature about these species. The health effects related to the species are described as well as the associated pathogens.

**Abstract:**

Members of the Cimicidae family are significant pests for mammals and birds, and they have attracted medical and veterinary interest. A number of recent studies have investigated bed bugs, due to their dramatic resurgence all over the world. Indeed, bed bugs are of significant public health and socioeconomic importance since they lead to financial burdens and dermatological complications and may have mental and psychological consequences. It is important to note that certain cimicids with a preference for specific hosts (birds and bats) use humans as an alternative host, and some cimicids have been reported to willingly feed on human blood. In addition, members of the Cimicidae family can lead to economic burdens and certain species are the vectors for pathogens responsible for diseases. Therefore, in this review, we aim to provide an update on the species within the Cimicidae family that have varying medical and veterinary impacts, including their distribution and their associated microorganisms. Various microbes have been documented in bed bugs and certain important pathogens have been experimentally documented to be passively transmitted by bed bugs, although no conclusive evidence has yet associated them with epidemiological outbreaks. Additionally, among the studied cimicids (bat bugs, chicken bugs, and swallow bugs), only the American swallow bug has been considered to be a vector of several arboviruses, although there is no proven evidence of transmission to humans or animals. Further studies are needed to elucidate the reason that certain species in the Cimicidae family cannot be biologically involved in transmission to humans or animals. Additional investigations are also required to better understand the roles of Cimicidae family members in the transmission of human pathogens in the field.

## 1. Introduction

The order of Hemiptera insects are known as “true bugs”, and they represent around 90,000 species of insects in the world [1,2]. Most hemipterans are either phytophagous or entomophagous and few hemipterans are hematophagous. The entirely bloodsucking heteropteran insects include, for example, the Polyctenidae family (vitally related to bats), the Reduvidae family (only subfamily Triatominae), and the Cimicidae family [3]. Members of these families range from small to large in size and, at all developmental stages, they are hematophagous [4,5]. The Polyctenidae family comprises 32 species that are obligate ectoparasites of microchiropteran bats, with no considered economic and medical importance [2]. These insects are wingless, viviparous with only three nymphal instars, and they also lack eyes and ocelli [2,3,5]. The subfamily Triatominae (Reduvidae family) comprises more than 110 species including diverse vectors or potential vectors of *Trypanosoma cruzi* that are responsible for American trypanosomiasis [6]. These insects feed on several vertebrates, mammals including humans, birds, and reptiles [4]. Kissing bugs or triatomines are of great medical and veterinary importance because they are vectors of *Trypanosoma cruzi*, the causative agent of Chagas disease [7]. This disease is considered to be a major medical problem in South America and is known to be fatal to humans [4,6]. The third hemipteran family is Cimicidae. Members of the Cimicidae family are ectoparasites of mammals, birds, and more rarely, reptiles. They have preferred hosts but they can feed on a range of hosts if their preferred host is absent [8,9]. The Cimicidae family consists of six subfamilies, namely Primicimicinae, Cimicinae, Cacodminae, Afrocimicinae, Latrocimicinae, and Haematosiphoninae, 24 genera, and over 100 species [9]. Cimicidae are small, wingless bugs that measure between 2.5 mm and 7 mm in their adult state; one species, *Primicimex cavernis*, which is considered to be one of the most primitive species, is large and measures between 10 and 13 mm. The bodies of these insects are flattened, from pale to dark brown in color, and sometimes reddish-brown. They are from rounded to oval in shape with varying quantities of bristles depending on the species. All the developmental stages feed on vertebrate blood [2,6,10]. The mode of reproduction in Cimicidae is unique. All species exhibit traumatic insemination. The phenomenon has been well described by Carayon [11]. The male’s clasper transpierces the female’s abdomen and injects the sperm. The point of puncture and deposition appears to be random in primitive cimicids compared to other cimicids including bed bugs (mesospermalege) [6].

Cimicids are temporary ectoparasites living separately from their hosts, but bat-associated bugs have been reported to depend strictly on their host for dispersal strategies [12,13]. When not feeding, they hide in cracks and crevices in caves, bird nests, and human dwellings [6]. The relationship between humans and these insects is very old, dating back to prehistoric times when, for instance, *Cimex* (*C.*) transitioned from living off bats to primitive humans [8,9]. Although several species can bite humans, two are strictly related to them: *Cimex lectularius* and *C. hemipterus*. Bat/bird bugs and poultry bugs are adventitious feeders on humans in the absence of their main hosts or when the population increases and spreads [9,14,15]. Globally, cimicids are not known to contribute to the development and transmission of disease agents to humans in natural settings. However, they have been associated with many pathogens [9,16]. Moreover, this variety of ectoparasites is of medical and veterinary interest. In this review, therefore, we focus on the human public health and veterinary impacts of the Cimicidae species. Our review aims to provide recent and detailed insight into the main species that infest humans as well as animals, the economic losses they can cause, their distribution, and their associated pathogens. In addition, we report on the laboratory studies that have investigated their potential vector competence of transmitting infectious agents.

## 2. List of Species Belonging to the Cimicidae Family of Medical and Veterinary Importance, Their Distribution and Their Public Health Impact

Among the six subfamilies of Cimicidae, some species from Cimicinae, Cacodminae, and Haematosiphoninae have minor or major medical and veterinary importance [1,6]. For example, in the Cimicinae subfamily, there are bed bugs (*C. lectularius* and *C. hemipterus*) that strictly infest humans; *Cimex vicarius* and *C. hirundinis* that feed primarily on swallows, i.e., house martin and cliff swallow, respectively [9,17]; *C. pilosellus*, *C. pipistrelli*, *C. insuetus*, and *C. adjunctus* that use bats as their primary hosts; and *C. columbarius* that infests exclusively pigeons [9,18]. Among the members of the Cacodminae subfamily, *Leptocimex boueti*, *Stricticimex parvus*, and *S. antennatus* have a preference for bats [18,19,20]. Finally, the Haematosiphoninae subfamily, including *Haematosiphon inodorus* and *Ornithocoris toledoi* regularly feed on poultry and *Cimexopsis nyctalis* is an ectoparasite of chimney swifts [9,15,21,22]. In this review, we describe the bugs in terms of the trophic level of their hosts. For example, mainly, Vespertillionidae and Molossidae are the preferred blood sources of bat bugs [8], and specific species of the family Hirundinidae are preferred by swallow bugs [9,23].

### 2.1. Bed Bugs

Bed bugs are primarily human ectoparasites that belong to the Cimicinae subfamily. They measure between 4 and 7 mm in adult form. There are two species, the common bed bug (*Cimex* (*C*.) *lectularius*) and the tropical bed bug (*C. hemipterus*) [9,24]. *Cimex hemipterus* lives in tropical, certain temperate countries and *C. lectularius* lives in most parts of the world [25]. The biological cycle includes eggs, five immature stages, and adults [9,26]. Their multi-stage developmental life cycle requires a blood meal every three to five days [16]. They can survive without feeding for over a year [27]. Minor contaminations by bed bugs can suddenly turn into acute infestations. Usinger [9] reported that experimental mating between *C. hemipterus* and *C. lectularius* occasionally produced first generation eggs but failed to produce second generation adults. In addition, crosses between *C. hemipterus* males and *C. lectularius* females were fatal and toxic to the latter. The two species were both identifiable only by entomologists, for instance, by measuring the pronotum width/length ratio and other diverse morphological criteria [9].

#### 2.1.1. *Cimex lectularius*

The common bed bug, *Cimex lectularius* (Figure 1(A1,A2)), is a cosmopolitan ectoparasite that lives in temperate or tropical regions, and has been reported in several countries in all continents except Antarctic (Table 1) [9]. This species is closely associated with humans but can bite bats, birds, reptiles, and, experimentally, amphibians [19]. Its parasitic relationship with humans has been document in ancient Egyptian sites [28], and it spread throughout the world in the 20th century. The association with birds and bats has been reported in Europe but has not been recorded in North America [29,30]. By occupying crevices and cracks in human dwellings, bats have transferred their cimicids into close contact with humans, which created conditions for cimicids to adapt to a new host [9]. Balvin et al. (2012) [29] also suggested a 20% higher size in European *C. lectularius* lineages close to humans compared with ancestral lineage mainly associated with bats. *C. lectularius* bites at night and is active all year round in very warm and temperate countries. The species is identified by different criteria such as a pronotum width/length ratio that is greater than 2.5 in *C. lectularius*, with large pronotal lateral lobes. The shape of the pronotum bristles is forked and sharpened [31]. The paragenital sinus is clefted and bristled [9].

#### 2.1.2. *Cimex hemipterus*

*Cimex hemipterus* (Figure 1(B1,B2)) is a human pest, which was originally located in tropical areas. The tropical bed bug is native to south and southeast Asia (Table 1). Before World War II, the species was exclusively recorded as being tropical. It was also reported in Africa and Australia [9,52]. The distribution of *C. hemipterus* has hugely expanded since then, moving beyond tropical and subtropical zones and into temperate countries. Surprisingly, this species has been found in the Donbass region, Russia, Sweden, France, and Italy [53,54,55,56,57]. *Cimex hemipterus* has adapted to humans but also occurs on chickens. It was originally an ectoparasite of bats such as *C. lectularius* [9]. The pronotum width/length ratio of *C. hemipterus* is less than 2.5 with no broadly expanded lateral margins. In addition, the pronotum bristles are smooth and not jagged [31].

#### 2.1.3. Medical Impact of Bed Bugs

In the literature, approximately 65 microorganisms are considered to be potentially transmissible by Cimicidae [52]. Most studies have been conducted on wild or laboratory colonies of Cimicidae. Bed bugs, including *C. lectularius* and *C. hemipterus*, are of great importance to public health. To date, there is no epidemiological evidence to incriminate human bed bugs as vectors of human and animal pathogens. However, experimental models around the world have shown that bed bugs can acquire, maintain, and transmit microorganisms, although no definitive conclusions have been drawn about the transmission power of these bugs, mainly in nature. However, more than 50 pathogenic microorganisms have been detected in bed bugs [32,52].

Contamination routes may include the salivary glands, although only one study [58] experimentally reported *Rickettsia parkeri* from *C. lectularius* salivary glands following a blood meal. Another route of contamination could be through feces such as for *Trypanosoma cruzi* [59], *Borrelia recurrentis*, or *Bartonella quintana*. It has also been proposed by Andrés Zorrilla-Vaca that crushing a bed bug during feeding could cause contamination, as is the case for body-louse bacteria [52]. Previous studies have shown that many infectious agents remain viable in bed bugs and could be transmissible under laboratory settings [52,60]. In this review, however, we provide an update of the most important pathogens.

With regard to Chagas disease, *T. cruzi* has been reported in mammals in North, Central, and South America, but not in amphibians or birds. Birds are refractory to this parasite but can act as feeding hosts for the triatomine bug [10]. Chagas disease is normally transmitted by kissing bugs. Bed bugs and kissing bugs share a particular behavior, namely reflexive feces excretion behavior after having a blood meal or during a blood meal for the best triatominae vectors (especially *Rhodnius prolixus* and *Triatoma infestans*) [16]. This behavioral feature in kissing bugs is the main cause of *T. cruzi* transmission [16]. Parasite penetration is eased by scratching pruritic bites. *Cimex hemipterus* and *C. lectularius* have been reported to acquire and maintain the parasite that is then detected in the feces. Transtadial transmission has also been reported [34,60]. In Peru, Salazar et al. (2015) concluded experimentally that the *T. cruzi* load in bed bug defecation was identical to that of the triatominae vectors and, surprisingly, *T. cruzi* transmission was significant and bidirectional [59]. It has been reported, in Argentina, that *C. lectularius* fed on *T. cruzi* wild-infected rodents were able to transmit the parasite and the transmission was as efficient as the triatomine bug [61]. The co-occurrence of bed bugs and triatomine bugs in the same environment is possible, but bed bugs in the wild do not normally excrete on their hosts. This behavior may lower the transmission of parasites [16,32]. Contrarily, Darrington (2015) [62] stated that bed bugs post-feeding defecation behavior might potentially facilitate the transmission. However, no epidemiological evidence has been reported and *T. cruzi* transmission in natural settings is currently lacking [63]. Further studies are needed to explore its transmission via bed bugs in the wild [32].

Blood-borne viruses such as hepatitis B virus (HBV) and human immunodeficiency virus (HIV) have gained much interest, and several investigations have been conducted to study the possible transmission of these viruses by bed bugs. Hepatitis B virus has been detected in both wild and laboratory bed bugs. Under laboratory conditions, it has persisted for up to 35 days in bed bugs and was also detected in feces as well as being maintained transtadially but not transovarially [64,65]. Otherwise, the biological replication of HBV has not been observed in infected blood-fed bed bugs, since viral loads have continuously decreased over time [16,60]. Another study was unable to confirm the transmission to chimpanzees [66]. Viral hepatitis has also been detected in bed bugs in India [67]. A study in Gambia [68] investigated the epidemiological link between the presence of bed bugs and HBV-infected children. The rate of infection remained stable even after the bed bugs had been eliminated. It was concluded that the bed bugs were not associated with childhood transmission of HBV. Another study in India reached the same conclusion [64]. Ultimately, the potential of HBV transmission through bed bugs is insignificant. A recent study identified HCV (hepatitis C virus) in *C. lectularius* recovered from human dwellings in Europe [69]. So far, there is no proof or solid evidence of human infection with HBV or HCV after being in contact with bed bugs [60].

Regarding the human immunodeficiency virus (HIV), it has never been detected in wild bed bugs [64]. An initial examination of HIV in *C. lectularius* showed that the virus survived in bed bugs for an hour after feeding on infected blood [70]. A similar observation was reported in a later study in which it was shown that HIV persisted in bed bugs up to eight days after experimental feeding. Furthermore, viral replication and virus detection in the feces of these bed bugs was not observed. Therefore, HIV transmission is unlikely to occur in the human environment and no bed bug-borne transmission has been confirmed [16]. Although the described viruses (HBV and HIV) can persist in bed bugs for a specific time, no infectivity and no viral multiplication has been observed, which points towards a minimal risk of bed bugs being vectors of these viruses [8].

In one previous study, five bed bugs were collected from an infested room in Vancouver, Canada. Methicillin *Staphylococcus aureus* (MRSA) was isolated from three bed bugs and the remaining two bed bugs were infected with vancomycin-resistant *Enterococcus faecium* (VRE) [71]. The authors hypothesized that the transmission to bed bugs might be due to feeding on infected blood, although it was not reported whether the residents had been infected with the two strains. Accordingly, Lowe and Romney (2011) [71] suggested that bed bugs could be vectors of MRSA. Nevertheless, this was refuted by another study showing experimentally that the bacteria did not survive in the midgut of *C. lectularius* for more than nine days after feeding on MRSA-infected blood [72]. In addition, they observed no amplification of MRSA in the midgut. For this reason, they concluded that bed bugs are unlikely to be active in the transmission of this bacteria [72]. Recently, a case was reported of a patient who sustained an *S. aureus* infective endocarditis and skin lesions caused by bed bug bites. However, there is no literature supporting the association of infective endocarditis caused by *S. aureus* and bed bug bites [73].

In a review, Delaunay et al. (2011) [16] reported on the possible transmission of *C. burnetii* through bed bugs. Transtadial transmission was demonstrated, the bacteria were isolated at all stages, and the infection was maintained up to 250 days without any amplification. Then, the bacteria were excreted in the feces of bed bugs. The potential ability of transmission to humans or animals does, therefore, need to be considered and confirmed, since *C. burnetii* is basically transmitted through aerosols and in other ways [60,74]. Delaunay et al. (2011) and other authors [8,16,75] also reviewed the potential transmission of fungi, since they have been detected in *C. lectularius*, but no evidence has proven its transmission to humans via bed bugs.

*Wolbachia* are transovarially inherited intracellular endosymbionts. They are naturally harbored in bed bug gonads [76,77]. This endosymbiont is not known to be transmitted either to humans or via blood feeding. *Wolbachia* is crucial for the growth and reproduction of bed bugs (male and female fitness) as they provide vitamin B biotin and riboflavin [78,79]. They may even protect bed bugs against parasitoids and pathogens. Compared to other arthropods, high temperatures could lead to the elimination of *Wolbachia* from bed bugs that negatively affect their fecundity and development [80,81]. Thus, bed bugs are unlikely to survive and maintain reproduction. The focus on Wolbachia for control purposes needs further research to better understand the interactions and relationships between the endosymbiont and bed bugs. Moreover, their associated endosymbionts have yet to be explored [82].

In contrast, Goddard et al. (2012) [58] experimentally fed bed bugs (species not mentioned) on *R. parkeri*-infected blood. They found the presence of the bacterium in the salivary glands at 15 days post-infection using immunofluorescence, although this could not be maintained transtadially. Thus, the transmission of *R. parkeri* is unlikely to occur naturally. Recently, uncharacterized *Rickettsia* was detected in *C. lectularius* field samples but its pathogenicity was not elucidated; therefore, further research is required to more fully determine this bacteria [83].

*Borrelia recurrentis* is transmitted to humans through contamination with infected feces on scratches, through crushing body lice, and through skin lesions [84]. Under laboratory conditions, bed bugs (*C. lectularius*) have been considered to be competent vectors of *B. recurrentis* since it remained viable even at 20 days post-infection and was excreted in the feces. A recent study showed that *B. recurrentis* could reach the hemolymph and could invade the major host barrier within 24 h, similar to body lice. In addition, they indicated that bed bugs were more likely to be crushed, similar to body lice. Nevertheless, the bacterium seemed not to be maintained permanently in the hemolymph [85]. Therefore, definitive research is needed since the transmission under natural conditions remains unclear [86].

*Burkholderia multivorans* is an opportunistic nosocomial bacterium. It is known to pose an important threat to immunocompromised individuals. It is also known to be resistant to antimicrobial agents [87]. The DNA of this pathogen was detected in several bed bugs collected from units in an elderly care facility in the United States [88]. A recent study demonstrated the acquisition of *B. multivorans* by bed bugs for up to 13 days post-ingestion, although the bacteria was undetectable at 16 days post-ingestion [89]. Moreover, the bacterium was not observed in the saliva or in the progeny by vertical transmission from infected bed bugs [89]. The ability of bed bugs to disseminate *B. multivorans* to humans or the environment, therefore, appears to be unlikely [89].

The DNA of the trench fever agent *Bartonella quintana* was detected in *C. hemipterus*, collected from Rwandan prisons where *B. quintana* and *R. prowazekii* had previously been identified in body lice. However, the authors could not isolate the bacterium by culture from bed bugs [90,91]. Subsequently, in an experimental study, *C. lectularius* were fed blood infected with *B. quintana*. *B. quintana* persisted up to 18 days, was transmitted vertically to the progeny, and was found viable in the bed bug feces. Consequently, the study highlighted bed bugs as competent vectors of *B. quintana* [92]. Transmission in nature requires further investigation.

The role of bed bugs in the transmission of plague has long been speculated [93]. In 1897, a human case of plague was reported as possibly being linked to bed bugs [94]. Subsequently, in 1910, plague-infected bed bugs were found in a camp heavily contaminated with plague where no fleas had been detected [95]. The author then managed to infect rats using the infected bed bugs. According to Pollitzer [96], during plague epidemics, bed bugs are frequently found to be infected, sometimes even in the beds of plague patients. Experiments carried out in the early 20th century showed that *Y. pestis* survived in the stomachs and feces for more than 100 days [93]. Nevertheless, its transmission via bed bugs in the field requires further investigation [93].

Recently, *Francisella tularensis*-like has been detected in wild bed bugs collected in a rural area in Madagascar, but its pathogenicity needs to be determined [97]. In addition to the most important microorganisms detailed in the review, *Francisella tularensis*, *Brucella melitensis*, *Leishmania donovani*, and *Salmonella typhi* have also been detected in bed bug feces [34,98]. Generally, bed bugs defecate after a blood meal and not during feeding which limits transmission of pathogens. The excrements are normally represented as dark fecal spots on mattresses or around hiding places [26]. Although some studies have found different microorganisms in bed bugs and other research studies have demonstrated their potential ability to transmit pathogens experimentally, there are no studies that have confirmed cases of human disease transmission in natural infestations [99]. Furthermore, Cimicidae co-infestations with other medically important haematophagous arthropods are possible. For instance, co-infestations of body lice and bed bugs have been reported in a patient in USA. No louse-borne bacteria were detected, although the author raised a question over the potential risk of bed bugs acquiring louse-borne pathogens and acting as a secondary vector under certain conditions [100].

Although bed bugs do not play a significant role in transmitting human pathogens, they are still considered to be medically important since they have both physical and mental impacts [1]. Their frequent unpleasant bites trigger skin lesions, itching, and blood loss (in severe cases leading to anaemia) when individuals are chronically infested [101]. These insects usually bite their hosts at night. During the blood feeds, the bites are painless but the skin reaction or inflammation is triggered a few hours later, subsequent to an immune response initiated by bed bug saliva proteins (vasodilatory particles, aggregation inhibitors, and anticoagulant factors) [16,26,102]. The bites generally occur along the arms and the legs on uncovered areas of the skin. Skin reactions are seen in between 30% and 90% of people [103]. The most frequent clinical presentations are pruritic, maculopapular, and erythematous lesions [26,33]. Initial skin reactions are shown as macular lesions (2–5 mm) [26,64], which then evolve in diameter. They present as either circular or ovoid wheals and, in some cases, the wheals reach 20 cm in diameter, leading to bullous eruptions and nodules in extreme cases [26,103,104]. The central punctum related to the bite is not always visible although it can be in some cases [26]. According to some authors, the bites are either in a zigzag formation or a straight line, including between three to five bites, presented in the pattern of “breakfast, lunch and dinner” [105]. Conversely, the pattern and distribution of the bites are not obvious or specific for all bed bug bites [26,32]. Chronic bed bug infestations can lead to a chronic skin reaction, dermatitis, allergic reaction, and severe anemia (iron deficiency), but bed bugs have rarely been stated to cause asthma or bronchospasms [101,106,107,108,109]. Mental and psychological alterations are strongly related to bed bugs. They cause discomfort, fear, stress, social isolation, anxiety, depression, delusional disorder, insomnia, nightmares, and even suicide in rare cases [10,110,111].

### 2.2. Bat Bugs

Bat bugs are common parasites of bats worldwide. Bats are the second largest mammalian order and are considered to be the ancestral hosts of cimicids [9,112,113]. However, humans and other vertebrates may be secondary hosts. Bats provide a convenient environment for ectoparasites that mostly hide in cracks in the areas where bats roost and where they digest their blood meals [9]. Cimicids have a restricted choice of hosts compared to other haematophagous insects and they mainly infest molossid and vespertilionid bats [8]. There are two subfamilies within Cimicidae (Cacodminae and Afrocimicinae) associated with Old World bats, as well as two others (Primicimicinae and Latrocimicinae) narrowly related to New World bats. After bats migration, cimicids can survive up to 1.5 years without a blood meal [114]. Certain species of bat bug groups may occasionally bite humans living near nesting and roosting sites or when their host is absent [1,9]. However, they are not well-adapted to feeding on humans. An infestation with bat bugs can easily be mistaken for a bed bug infestation because macroscopically they appear identical to bed bugs and the differences are only noticeable microscopically [115]. Generally, bat bugs are distinguishable from bed bugs by having longer bristles over their whole body. The bristles on the pronotum are equal or longer than eye width compared to bed bugs and they have wider hind femurs than bed bugs [115].

#### 2.2.1. *Cimex pilosellus*

This species is exclusively found in North America (Table 1), mainly in western Canada and in western USA, in other words, from British Columbia to California and Arizona [41]. This is why this species is referred to as the western bat bug. It can be found in both urban and campestral settings depending on the bat hosts [116]. This species mainly feeds on several species of bats belonging to the genera *Antrozous*, *Eptesicus*, *Lasionycteris*, *Myotis*, and *Pipistrellus*, and occasionally or accidentally, they feed on humans, as they adopt similar behaviors to bed bugs by living in dark cracks and feeding on people at night [9,117]. All Cimicidae have identical morphologies; therefore, a human infestation with *C. pilosellus* is often confused with a bed bug infestation [41,118,119]. This species is morphologically characterized by longer and narrower hemelytral pads (width ratio = 1.6 or 1.7) [9].

#### 2.2.2. *Cimex pipistrelli*

*Cimex pipistrelli* (Figure 1D) is a typical ectoparasite of bats. It has a wide range of distribution in the Palaearctic region (Table 1) [120]. It is a very common ectoparasite found in the shelters of many European bat species. This species can co-habit without physical contact with bats [8]. They inhabit crevices in close proximity to their bat hosts. In general, they prefer a lower roosting temperature than bats, because the temperature of the host is too high for their development [121]. Therefore, they target bats only for a short time to feed and are not normally found in the fur. If the bat hosts are absent for too long and they are near households, they may move into human dwellings to feed [42]. Morphologically, this species has very long bristles at the sides of the pronotum. The bristles are longer than the width of the first antennal segment (more than 0.13 mm) and the bristles of abdominal tergites are longer than the distance between bristles [9]. The *Cimex pipistrelli* group is distinguished from other groups of Cimicinae by having narrow pronotal lateral lobes and cleft and naked paragenital sinus [9].

#### 2.2.3. *Cimex adjunctus*

*Cimex adjunctus* is a widespread ectoparasite of different species of bats in North America, especially in the eastern United States (Table 1) [122]. It has also been observed in temperate regions and tropical areas [18]. This species parasitizes on several insectivorous bat species [9,123]. *Cimex adjunctus* occasionally feed on human blood when their preferred bat host is not available. It is known to move to occupied rooms and bite people during the night. This species is morphologically characterized by having long and thin pronotal bristles (0.2 mm) and long bristles on the hind tibia which may be as long as the width of the tibia, over 0.9 mm [9].

#### 2.2.4. *Stricticimex parvus* and *Cimex insuetus*

These two species inhabit caves close to bats. *Stricticimex parvus* has been collected from caves in Thailand, whereas *C. insuetus* is geographically extensive and has been found in Thailand, China, and India (Table 1) [18,46]. The bat host species is likely to be *Tadarida plicata*. However, the specific bat host has not been definitively identified between the two species *S. parvus* and *C. insuetus*, as different bat species have been found in the caves. These bugs have been reported to attack humans ferociously when they enter caves, including when local workers temporarily reside within bat-dwelling caves to collect guano [46]. Morphologically, *S. parvus* is particularly similar to *C. insuetus* by being pale in color, having long bristles, and a developed pronotum [124]. Nevertheless, in *S. parvus*, the interocular space is as wide as the length of the second antennal segment and shorter than the second antennal segment in *C. insuetus* [124].

#### 2.2.5. *Stricticimex antennatus*

This species occurs in South Africa and has been found and observed in an abandoned mine, living in bat roosts. The species of bats present here were *Rhinolophus simulator* and *clivosus*, *Myotis tricolor*, *Hipposideros caffer*, and *Nycteris thebaica*. *Stricticimex antennatus* has been observed feeding on the wings or forearms of these bats, and people entering an abandoned mine adit (Doornhoek Cave) in South Africa have been bitten several times [20]. Morphologically, the species is differentiated by being large in size, the pronotum 1 mm wide or more and the third antennal segment more than twice as long as fourth [9].

#### 2.2.6. *Leptocimex boueti*

*Leptocimex boueti* (Figure 1E) is between 3 and 4 mm in size. This species was first discovered in Guinea. It is very common in the Ivory Coast, specifically in Gaoua [19]. This West African cimicid has been recorded in cave bats. It is also known as a human ectoparasite and has been documented to infest human dwellings in rural regions. Morphologically, it is easily recognized by its long legs and the pubescence of the body. *Leptocimex boueti* is distinguished by having a distinct longitudinal row of bristles (15–20 bristles) on the inner posterior face and two rows of bristles on the ventral side of the front femora, as well as by its elongated third antennal segment, adaptive long legs for living in caves, and its thorax that is not much wider than its head [9,47].

#### 2.2.7. Medical and Veterinary Importance of Bat Bugs

With regard to bat bugs, a non-pathogenic trypanosome has been found in the gut of *C. pipistrelli* collected from the field in Europe [18]. Later, another study [43] demonstrated that *Trypanosoma incertum* was able to develop in laboratory-reared *C. pipistrelli*. The gut content of the infected bugs was then recovered and injected into six uninfected bats (*Pipistrellus pipistrellus*). The uninfected bats all later tested positive for *Trypanosoma*. The Kaeng Khoi (KK) virus was first isolated in central Thailand from bat species. It was then detected in *S. parvus* and *C. insuetus* which live together in caves [46]. These two species fiercely attack people who enter the caves to collect guano. Specific antibodies to KK have been found in these people but the symptoms of the virus in bats and human remain unknown. Nevertheless, the guano collectors believed that the bites from these bugs were responsible for influenza-like illnesses including high fever and joint pain [46]. The virus could be of concern to public health, since 29% of the bat guano collectors tested seropositive for KK virus, although its pathogenic potential is still unclear [125,126]. Williams et al. (1976) [46] claimed that further laboratory studies were needed to evaluate the ability of *S. parvus* and *C. insuetus* for KK virus transmission. *Stricticimex antennatus* has repeatedly bitten people visiting an abandoned mine in South Africa [20]. The bite was painless but later the area became reddened, swollen, and itchy, and some reactions included pustule formation [20]. A novel *Bartonella* has been associated with wild *C. adjuntus*, whereas *C. pilosellus* has been reported to transmit *T. cruzi* in laboratory conditions [18,44]. In addition, Brumpt (1912) [19] reported that *L. boueti* allowed the evolution of *T. cruzi* and it was considered to be a bed bug because it willingly feeds on humans [19].

### 2.3. Bird Bugs

#### 2.3.1. Swallow Bugs

Swallow bugs are very small in size, measuring between 2.5 and 3.7 mm [17]. These insects are ectoparasites of birds, mainly swallows. In the absence of their primary host (i.e., bird), swallow bugs can survive for a long period of time without feeding and they can leave their nests and feed on the blood of alternative nearby hosts, including humans [17]. These insects belong to the Cimicinae subfamily and the genus *Oeciacus* but were transferred to the genus *Cimex* by Balvin et al. (2015) [127]; in this review, we follow the suggestion of *Cimex*, based on previous recent publications [31,128]. The bodies of swallow bugs are covered with pale, thin and long bristles and their pronotum is moderately developed. They are also characterized by thicker third and fourth antennal segments [9]. Three species infest swallows: *C. hirundinis*, *C. vicarius*, and *C. montandoni*.

##### *Cimex hirundinis* 

*Cimex hirundinis* (Figure 1C1,C2) is a species with European distribution, that lives in western and central Europe as well as the Mediterranean basin (Table 1) [9]. It is primarily an ectoparasite of the house martin (*Delichon urbica*) [129], and is less commonly found in other Hirundinidae. It has been recorded in at least 16 birds [23]. Breeding of this species sometimes occurs in swallow nests. The featherless new-born birds are their potential target and they can die as the result of multiple bites [17]. The insect overwinters in the nests as an adult or nymph, apparently remaining without blood meals during the long months when their hosts are absent. Extreme resistance to hunger and cold has also been demonstrated [17]. Human infestations with *C. hirundinis* have recently been reported in France [31], Italy [130], and Japan [131]. Their bites are described as very painful, causing an accentuated and persistent swelling [17,19]. Usinger (1966) [9] reported that crosses between *C. hirundinis* from Greece and *C. vicarius* from the USA (California) were entirely intersterile. The *Cimex hirundinis* morphology is characterized by being small and the pronotum width/length ratio is not less than 2.5. The body is covered with long, thin bristles and the rostum and antenna are short. The lateral pronotal lobes are less developed [9,17].

##### *Cimex vicarius* 

The American swallow bug, *C. vicarius*, is located in North America (mostly in the transmontane western United States) and south to Durango Mexico [1,9]. *Cimex vicarius* infestations occur primarily in the nests of cliff swallows (*Petrochelidon albifrons*) and are rarely found in the more open nests of the barn swallow (*Hirundo erythrogaster*) [9]. *C. vicarius* can survive in abandoned swallows’ nests several months without feeding but not more than a year and some may have alternative hosts nearby such as bats, other birds, and in some cases, humans [132]. The biology of this parasite was studied by Myers (1928) [133] who reported that the eggs of females fed on human blood did not reach maturity as well as when fed on swallow blood. In nature, these insects are attacked by the spider *Latrodectus mactans* [19]. The American swallow bug differs from the European swallow bug (*C. hirundinis*) by being larger, having a light color, and by the structure of the antennae. The anterior edge of the prothorax is also slightly curved. The pronotum width/length ratio is less than 2.3 and the abdomen bristles are very long [9].

##### Medical and Veterinary Impact of Swallow Bugs

The American swallow bug (*C. vicarius*) is known to be a vector of the Buggy Creek virus (BCRV) [39] which is a strain related to a western equine encephalomyelitis virus (WEE virus), and to transmit a Tonate virus (“Bijou bridge”) related to Venezuelan equine encephalitis (VEE) [36]. The Stone Lakes virus (a variant of Fort Morgan virus, FMV) has also been isolated from swallow bugs [37]. The first isolation of Buggy Creek virus, related to WEE virus, was from *Cimex vicarius* [35]. It was then repeatedly isolated from naturally infected cliff swallow bugs and their hosts (the cliff swallow and house sparrow). A potential vertical transmission of BCRV to eggs collected in the wild from *C. vicarius* has also been reported [38]. Another study demonstrated the persistence of this virus for two years in unfed *C. vicarius* samples collected from nests left vacant by cliff swallows [39].

*Fort Morgan* virus has often been isolated from cliff swallows and their bugs, and house sparrows [35]. In another study, field-collected *C. vicarius*, which were fed on infected house sparrows with FMV, showed that they were infected by up to 70% to 80% after 18 days of incubation under laboratory conditions. These swallow bugs were subsequently fed on uninfected birds. The data showed that *C. vicarius* were able to transmit the virus to the uninfected birds and they served as a long-term reservoir [134]. The American swallow bug acts, in general, as a vector and a reservoir for swallows, but there is still no evidence of this in humans [35,36,132,134]. *Cimex vicarius* is unlikely to be a competent vector of West Nile virus, as attempts to transmit it experimentally have failed. The virus was not amplified and cleared up at 15 days post-exposure [135].

In contrast, European swallow bugs (*C. hirundinis*) have been reported to serve as experimental host vectors for *Trypanosoma cruzi* [19,34]. One study conducted on *C. hirundinis* collected from nests in Slovakia revealed that these bugs were potential vectors of paramyxovirus type IV, since the infection rates in adults were 0.1% and 0.4% in II and V nymphs (transtadial transmission), respectively [40]. Very recently, a new *Wolbachia* named *Wolbachia massiliensis* was detected in *C. hirundinis* in France [31]. This species has never before been stated to be a vector of human pathogens. The swallow bug species may serve as an alternative host when their preferred host is absent [136]. A case of occupational human infestation with *C. hirundinis* in Italy showed that two farm workers sustained papules covered with haemorrhagic crusts on their abdomens and trunks [130]. The same lesions had been observed in the winter season for the previous 5 years. During winters, swallows were absent, and inspection of the inhabited nests revealed that they were mainly infested with immature stages of *C. hirundinis*. This case was likely due to the long fasting of the insects which caused them to feed on the blood of atypical hosts such as humans [130]. Human infestations with *C. hirundinis* have also been reported in France and Japan [31,131]. Moreover, the co-occurrence of a hospital infestation with two hematophagous arthropods, *Cimex vicarius* and *Argas cooleyi* has been reported in Tucson, Arizona, USA [136]. *Cimex vicarius* and *C. hirundinis* are also pests of considerable importance to swallows because they cause nestling mortality due to severe blood loss when infestations are heavy [1].

#### 2.3.2. Poultry Bugs

Poultry bugs principally infest chickens, and other poultry, and may accidentally infest humans. During the day, these insects hide in cracks and crevices but they are active at night around poultry roosting areas [48,51,137]. Human bites have also been reported during the night when activities occur close to poultry roosting areas. *Haematosiphon inodorus* and *Ornithocoris toledoi* are the main chicken pests of veterinary importance because they can be fatal to poultry and can lead to serious economic burdens in commercial poultry settings [1,15,48]. 

##### *Haematosiphon inodorus* 

*Haematosiphon inodorus* (Figure 1F) is also known as the Mexican chicken bug. It is smaller in size (3.34 mm) than *C. lectularius* but identical in size to *C. vicarius* [48]. It was principally first found in Mexico, followed by Central America, but it has also been found, more rarely, in the southwestern United States (Table 1) [15]. This species has been reported in only nine species of birds, including chickens, turkeys, owls, golden eagles, and the Californian condor [21]. The first collections of Usinger’s *H. inodorus* from native hosts were sampled from the great horned owl, the Californian condor, and two other non-identified owls [9]. Usinger [9] suggested that chicken infestations with *H. inodorus* may occur when condors, owls, and eagles are preying on chickens, and drop these bugs into chicken coops [9]. Human infestations are rare, but the Mexican chicken bug shows no reluctance to feeding on humans [48]. Poultry farmers have complained of bites from *H. inodorus* [50], and Townsend [138] reported the spread of this species from roosts to human dwellings. In addition, another case has been reported in Arizona (30 miles east of Kingman) where this species invaded a schoolhouse and bit children [139]. *Haematosiphon inodorus* has exceptionally four haematophagous nymphal instars compared to other cimicids [48]. Morphologically, it is distinguished by having a long rostrum, broad head, and a dorsal Berlese’s organ [48].

##### *Ornithocoris toledoi* 

*Ornithocoris toledoi* is between 3.94 and 4.20 mm in size [9]. This species has only been found in chickens, mainly *Gallus domesticus* in São Paulo, where it was first described [140]. This Brazilian chicken bug has also been found in Argentina and Bolivia in South America (Table 1) [9]. This species reportedly causes weakness to chickens due to blood loss and can result in the death of young chickens in the event of a heavy infestation [141]. *Ornithocoris* sp. and *O. toledoi* have been reported to enter dwellings at night and disturb residents. It has been suggested that it feeds on human blood in the absence of its preferred hosts [15,51]. *Ornithocoris pallidus* has also been reported to colonize human dwellings in the United States (Louisiana), but has mostly been described in blue and white swallows in Brazil [9,142]. The identification of *O. toledoi* is based on the characteristics of the pronotum being wider than long (more than 1.8 times), the third and fourth segment of the antenna being narrower than the second, the presence of two pairs of bristles in the posterior prothoracic angles, as well as the presence (in males) of two tufts in the final third of tibia I and II, but only in tibia II in females [9,137].

##### Impact of Poultry Bugs

*Haematosiphon inodorus* and *O. toledoi* can have a huge impact on commercial poultry farming. The nest boxes of broiler breeders can easily provide a shelter for poultry bugs. Chicken bugs can trigger skin lesions, breast and leg irritation, significantly reduced egg production, weight loss, stress, anemia, and morbidity in cases of heavy infestation and nestling mortality [1,143]. Chicken bugs also willingly attack humans when the nests of their natural hosts (domestic poultry) are in the vicinity of human dwellings [1,21,137]. *Ornithocoris* spp. has been stated to cause skin lesions and pruritic lesions in humans [137]. Similarly, *Haematosiphon inodorus* has been responsible for cases of dermatitis and pruritus in humans [50]. Poultry bugs have never been involved in the transmission of avian pathogens but they do cause a drop in egg production, and hence, lead to economic burdens [1]. Meanwhile, *H. inodorus* has been mentioned to allow the evolution of *T. cruzi* in laboratory settings. The species has been infected experimentally with *T. cruzi*, and 21 days post-infection, two white mice were infected by inoculating the bugs’ intestinal contents [34,49]. *Haematosiphon inodorus* is not known to be a vector of any infectious agent [1] although further study is necessary to assess their potential role in transmitting pathogens.

#### 2.3.3. Chimney Swift Bug

*Cimexopsis nyctalis* is found in the eastern part of USA, and it is an ectoparasite of chimney swifts *Chaetura pelagica* (L.). Welch (1990) [22] reported three cases of bites on people in Connecticut, and Schaefer (2000) [6] designated this species as an accidental biter of humans. This species is characterized by short, fine bristles on the dorsum and long bristles on the ventral surface. The pronotum is about twice as wide as long and the spines of tibiae are robust, almost as long as thickness of tibiae [9].

#### 2.3.4. Pigeon Bug

*Cimex columbarius* (Figure 1G) is in western Europe. It exclusively parasitizes and feeds off pigeons, possibly having adapted to pigeons because they have never been found in the cave nests of rock pigeons (Table 1) [9]. It also feeds on fly catchers (*Muscicapa atricapilla*). Morphologically, *C. columbarius* is closely related to *C. lectularius* but is differentiated by the size (smaller than *C. lectularius*) and shape of the prothorax. The ratio of the head width to the third antennal segment is 1.78 [14]. The taxonomy of the pigeon bug is still under debate; some authors have proposed it as a subspecies of *C. lectularius*, while others consider it to be a species in its own right [18]. The occurrence of hybridization between *C. lectularius* and *C. columbarius* is possible, but subsequent progeny appear to have low fertility [144]. *Cimex columbarius* has never been documented to bite humans [144]. In June 1924, a case of infestations of village dwellings with *C. columbarius* was reported in Amsterdam. The insect had nested under the roof tiles. Morphological examination revealed the species concerned was *C. columbarius*. It was suggested that the frequent bug infestations of many homes in Amsterdam may be due to pigeons nesting in attics [45].

## 3. Conclusions and Perspectives

Cimicidae are a major public health concern. This review summarizes and updates the cimicid species of medical and veterinary interest and highlights their associated microorganisms. Apart from usual human infestations with bed bugs, other cases of human infestations with birds, poultry, and bat bugs have been reported. Such cases are occasional and occur when the nests of either bats, birds, or poultry are in the vicinity of human habitations. These infestations can be confused with bed bug infestations. Therefore, it is important to confirm the origin of the infestation by investigating the surrounding dwellings. Previous studies have reported no conclusive evidence that significant disease outbreaks have been related to bed bugs or other cimicids. Various microorganisms including human pathogens have been detected in Cimicidae, although few laboratory experiments have demonstrated the competence of bed bugs to transmit important pathogens such as lice-borne bacteria. Conversely, no studies have shown their ability to maintain, amplify, or transmit these infectious agents in nature to either human or non-human hosts (birds, poultry, and bats). On the other hand, the American swallow bug has been considered to be a vector of several arboviruses, although there is no proven evidence of transmission to humans or animals. To date, no probable hypothesis has shown why bed bugs or certain Cimicidae are unable to transmit human pathogens. The most frequent hypothesis suggested is that bed bugs may have neutralizing factors that decrease virulence [99,145], but further studies are needed to elucidate the reason that certain Cimicidae cannot be biologically involved in transmission to humans or animals. Additional studies are also required to better understand the roles of Cimicidae family members in the transmission of human pathogens in the field.

## Figures and Tables

**Figure 1 insects-14-00392-f001:**
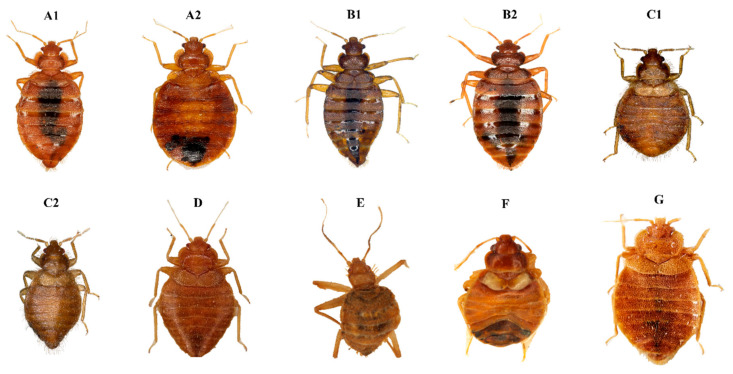
Habitus of main important cimicids: *C. lectularius* male (**A1**) and female
(**A2**); *C. hemipterus* male (**B1**) and female (**B2**);
*C. hirundinis* female (**C1**), male (**C2**); *C. pipistrelli*
female (**D**); *C. columbarius* female (**G**); *L. boueti* not specified
(**E**); *H. inodorus* female (**F**).

**Table 1 insects-14-00392-t001:** Summary of Cimicidae of medical and veterinary interest.

Subfamily	Species	Main/Secondary Hosts	Distribution	Medical and Veterinary Detected Pathogens	Natural or Laboratory Condition	References
Cimicinae	*Cimex lectularius*	Humans/bats, chickens and pigeons	Worldwide	- More than 50 microorganisms including viruses, bacteria, protozoa - Dermatological complications - Mental and psychological alterations	Naturally/experimentally	[26,32,33,34]
	*Cimex hemipterus*	Humans/bats and chickens	Tropical regions mainly Southeast Asia, Africa, Australia and Europe			
	*Cimex hirundinis*	Swallows/humans	Western and Central Europe. North Africa	- Possible vector of paramyxovirus type 4 - Experimental transmission of *Trypanosoma cruzi* - Dermatological manifestations	Naturally/experimentally	[1,9,19,34,35,36,37,38,39,40]
	*Cimex vicarius*		North America south to Durango in Mexico	- Transmission of several arboviruses - Vector of Buggy Creek virus - Dermatological manifestations		
	*Cimex pilosellus*	Bats/humans	Nearctic/North America	- Experimental transmission of *Trypanosoma cruzi* - Dermatological manifestations	Experimentally	[9,18,41]
	*Cimex pipistrelli*		Palearctic/Europe	- Experimental transmission of *Trypanosoma incertum* - Dermatological manifestations	Experimentally	[18,42,43]
	*Cimex adjunctus*		Nearctic/North America	- *Bartonella* spp. detection - Dermatological manifestations	Naturally	[44]
	*Cimex columbarius*	Pigeons/ humans (very rare)	Western Europe	- Dermatological manifestations: one case of human infestation	-	[45]
	*Cimex insuetus*	Bats/humans	Thailand, China and India	- Potential transmission of the Kaeng Khoi virus - Dermatological manifestations	Naturally	[46]
Cacodminae	*Leptocimex boueti*	Bats/humans	Tropical West Africa	- Evolution of *Trypanosoma cruzi* - Dermatological manifestations	Experimentally	[9,19,47]
	*Stricticimex parvus*	Bats/humans	Thailand	- Suspected transmission of the Kaeng Khoi virus - Dermatological manifestations	Naturally	[46]
	*Stricticimex* *antennatus*	Bats/humans	South Africa	- Dermatological manifestations	Naturally	[20]
Haematosiphoninae	*Haematosiphon* *inodorus*	Chickens/turkeys, humans, birds: owls, eagles, condors	Central America: Mexico, rarely in the southwestern United States	- Evolution of *Trypanosoma cruzi* - Skin lesion, anaemia and nestling mortality - Dermatological manifestations: Pruritus and dermatitis	Experimentally	[15,48,49,50]
	*Ornithocoris toledoi*	Chickens/humans	Brazil and certain South American countries	- Chickens: drop in egg production, skin lesions, weakness and anaemia - Dermatological manifestations: skin - Lesions, pruritic lesions	-	[1,15,51]
	*Cimexopsis nyctalis*	Chimney swift/ humans	North America (Eastern United States)	- Dermatological manifestations	Naturally	[22]

## Data Availability

All references used to write this review are listed and accessible.

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
