# Peer review of "Cimicids of Medical and Veterinary Importance"

_insects, 2023, doi:10.3390/insects14040392_

Round 1

Reviewer 2 Report

Peer review report on the manuscript "Cimicids of medical and veterinary importance", (Manuscript ID: insects-2236442).

Recommentation: Accept

Comments to Authors:

This manuscript is a review paper summarising the results of a long list of studies and other bibliographical references for the insects of the Cimicidae family regarding their world distribution, biology, parasitic behaviour, and their medical and veterinary importance.

The paper is well-written with a well-organized text, and the results of the literature review are clearly presented and adequately supported by the respective references. Thus, the paper compiles and provides important knowledge which is valuable for a better understanding of the medical and veterinary importance of the Cimicids.

For these reasons, the paper makes a substantial contribution to the literature and is therefore recommended for publication in the Insects, taking into consideration the following minor specific comments.

General comment: There appear to be many missing spaces between words or sentences in the way the PDF file of the manuscript is displayed. This fault occurs in almost all of the text and definitely needs to be fixed before the final version is released.

Furthermore,

The text in lines 28-32 is written in a smaller font size than the fonts in the rest of the text.

In line 56, delete one of the two full stops at the end of the sentence.

In Figure 1, consider writing the full name of the genus for the species that have not previously been shown in the text.

In line 168, which exactly is the bibliographic reference of Darrington 2015?

In lines 187 and 191, please provide the full name of the abbreviations HCV and HIV, as these are introduced for the first time in the text.

In the paragraph of lines 380 to 394, as well as in the paragraph of lines 500 to 514, the scientific names of species should be italicized.

In the Reference section, the scientific names of the species should also be italicized.

In Reference 6, the “Annu Rev Entomol” is repeated twice.

In Reference 25, change “cimex lectularius” to “Cimex lectularius”.

In References 58 and 84, please correct the names of the authors.

In Reference 111, please provide more details for the reference (and where it can be found) and write the correct names of the authors.

In References 124 and 127, please correct the names of the authors.

In Reference 129, please provide more details for the reference or consider excluding it from the reference list. Moreover, lowercase the words of the article to conform to the style for references suggested by Insects journal.

In Reference 131, please write the author's name in lowercase to conform to the style for references suggested by Insects journal.

In References 132 and 137, please correct the names of the authors.

Reviewer 3 Report

Suggestions and comments to manuscript “Cimicids of medical and veterinary importance”

This is a very interesting MS. The three authors present a very complete and exhaustive review of available published information on the importance of Cimicidae as vectors/transmitters of different types of pathogens for humans and different animal species.

I think this manuscript well done deserves to be published in Insects. However, before publishing, some minor issues need to be fixed.

♠ It is necessary that an expert polish the use of English.

♠ Avoid writing “Human” and “Bed bug” beginning with capital letters

♠ Lines 35-45. Remove the word “family” before each family. The ending “dae” denotes family

Remember to write each scientific name beginning with a capital letter and in italics. Pay special attention to lines 380-394, lines 500-513, and to the list of references.

♠ Lines 406-407. It is confusing to abbreviate scientific names that start with the same letter. Use the first two letters of each gender to avoid those problems

♠ Lines 45 and 49. The information is redundant.

♠ Lines 50-51. “…and is known to be fatal to humans [4,5]”. That is not true. Please include “potentially” before “fatal”.

♠ Line 55. “…but not in amphibians or birds.” That is inaccurate. Please consult:  10.1016/j.meegid.2022.105239

♠ Be careful of adding “sp.” or “spp.” after a genus, when appropriate.

♠ Table 1. Please include "Experimental transmission of Trypanosoma cruzi" for Cimex lecticularius.

♠ Reference 8. Usinger RL. Monograph of Cimicidae (Hemiptera-Heteroptera). Annap Entomol Soc Am. 1966. p. 585. This is a book, so change this format to the correct one.

♠ Reference 20. Pascal Delaunaye. The first name of this author should be abbreviated and in the correct place.

♠ Reference 33. Remove ….“No longer published by Elsevier”

♠ Reference 66. “Med Vet Entomol.” is repeated.

♠ Reference 78. “C Can Med Assoc J. Canadian Medical Association”. Remove the first four words.

♠ Reference 105. “M. Gresíková”. Interchange “places”, since last name should be in the first place.
